# Birds Eye View Look-Up Table Estimation with Semantic Segmentation

**Dongkyu Lee** [1], **Wee Peng Tay** [2] **and Seok-Cheol Kee** [3],*

1. Department of Smart Car Engineering, Chungbuk National University, Seowon-gu, Chungdae-ro 1, Cheongju-si 28644, Korea; dlehdrb3909@chungbuk.ac.kr
2. School of Electrical and Electronic Engineering, Nanyang Technological University, 50 Nanyang Ave, Singapore 639798, Singapore; wptay@ntu.edu.sg
3. Department of Intelligent Systems and Robotics, Chungbuk National University, Seowon-gu, Chungdae-ro 1, Cheongju-si 28644, Korea
* Correspondence: sckee@chungbuk.ac.kr

**Abstract:** In this work, a study was carried out to estimate a look-up table (LUT) that converts a camera image plane to a birds eye view (BEV) plane using a single camera. The traditional camera pose estimation fields require high costs in researching and manufacturing autonomous vehicles for the future and may require pre-configured infra. This paper proposes an autonomous vehicle driving camera calibration system that is low cost and utilizes low infra. A network that outputs an image in the form of an LUT that converts the image into a BEV by estimating the camera pose under urban road driving conditions using a single camera was studied. We propose a network that predicts human-like poses from a single image. We collected synthetic data using a simulator, made BEV and LUT as ground truth, and utilized the proposed network and ground truth to train pose estimation function. In the progress, it predicts the pose by deciphering the semantic segmentation feature and increases its performance by attaching a layer that handles the overall direction of the network. The network outputs camera angle (roll/pitch/yaw) on the 3D coordinate system so that the user can monitor learning. Since the network's output is a LUT, there is no need for additional calculation, and real-time performance is improved.

**Keywords:** birds eye view (BEV); look-up table (LUT); camera calibration; pose estimation; autonomous vehicle; MORAI Sim Standard

## 1. Introduction

Autonomous driving is currently receiving much attention, with many research institutes and companies conducting related research. The autonomous vehicle field can be divided into three major categories: recognition, judgment, and control. In particular, the cognitive area is rapidly developing, along with the development of machine learning. In addition, estimating 3D data based on 2D cognitive data using a camera is an essential element for autonomous driving. Estimating the pose of a camera attached to an autonomous vehicle is a method for outputting 3D data. In this paper, we propose a network that predicts an LUT [1] that transforms a camera image plane to a BEV [2] plane, and we aim to estimate the pose of the camera through this.

A camera is a sensor based on 2D data projected on a lens, so it is almost impossible to estimate perfect 3D data. Using a single camera to obtain the distance to an object—that is, the depth—is possible only with non-occlusion data, and the texture of the object with the Z-value cannot be accurately obtained. Therefore, we considered that a relatively strong feature point that can estimate a pose in the image plane is the free space and set, and we studied pose estimation based on the feature point in the area specified as free space as an initial goal. A semantic segmentation network [3–6] was selected as a backbone network

to construct a network that outputs the LUT by utilizing the feature points of the free space domain.

A BEV was used to show depth using camera data intuitively. This BEV is similar to the around view monitoring (AVM) [7] used for parking assistance [8]. However, since an object in a position higher than the ground is projected onto the floor and expressed, any obstacle on the road surface is expressed as drooping compared to the original shape. Moreover, since camera-based depth lacks a reference distance, the relative distance is valid, but it cannot calculate the absolute distance. In general, the ToF [9] sensor, the size of the surrounding artificial landmark, and motion data of a driving vehicle are used to estimate actual distance. However, the purpose of this paper was to obtain the camera pose of a driving vehicle by using only a camera, which is a low-cost sensor, so it does not matter that the only output is the relative distance when using a BEV.

Traditionally, to measure the pose of a camera, sensor calibration with a ToF sensor is performed, or the camera calibration process is performed in a tolerance calibration room [10], where a marker with a specified actual distance is located. However, the development of autonomous driving is progressing with the target of mass production, and it is necessary to set the cost of the sensor not too high so as to mass produce it. Therefore, we propose a system based on a single camera requiring other sensors and infra.

The rest of this paper is organized as follows: Section 2 introduces related works; Section 3 discusses the DB used in the experiment; Section 4 describes the structure and details of the entire network utilized in machine learning; Section 5 presents experiments with different methods and their results; finally, Section 6 concludes the thesis and presents future work.

## 2. Related Work

Le et al. [11] claimed that dynamic object detection and pose estimation are tightly coupled tasks. When a network is constructed and trained to perform dynamic object detection and pose estimation, the results of dynamic object detection and pose estimation work complementary. By applying this point, we used semantic segmentation that expresses the contour of an object rather than detection based on a bounding box. The LUT was predicted using the segmentation result as a feature point, and we constructed the network to estimate the roll/pitch/yaw of the camera image and to induce an interaction between the segmentation and pose estimation.

Jaderberg et al. [12] proposed STN, which continues the process of warping the original image and proposes a layer that can obtain an image that includes good features in contrast to the original image. In this process, a fully connected layer was placed inside the STN to consider the interrelationship of all features. We deduced the entire network to be suitable for pose estimation, to create a layer that outputs pose by itself, and to afford this layer the ability to convert encoded features into poses by using the fully connected layer.

Ronneberger et al. [13] proposed a general encoder–decoder and described how to efficiently perform up-sampling after down-sampling. Many studies have utilized similar methods. Our paper also deals with the form of an image to image (original image to the LUT) as a result, and since encoded data were used in processing it, the overall configuration was composed of an encoder–decoder.

## 3. Synthetic Database

We needed to collect image semantic segmentation and camera pose ground truth to implement the proposed network. Representatively, MS-COCO [14] and KITTI [15] have such data, but there is a disadvantage that the variation in camera poses is not significant. Each of the camera rolls, pitches, and yaws range from 0 to 360 degrees, but the variety of the open dataset has a disadvantage in that it does not reach that level.

Our solution was to use a simulator that simulates real environments and places. We acquired data from various camera poses by using the MORAI Sim Standard [16] (Figure 1) as a simulator. When a camera is attached to an actual vehicle, experimental data of various

angles cannot be acquired due to in-vehicle structures such as windshields. As learning proceeds, the results may not be generalized and may be overfitted. Since this paper proposes a network that predicts the LUT for producing a BEV using pose data, a simulator was used to derive data of the various poses that were constructed and utilized.

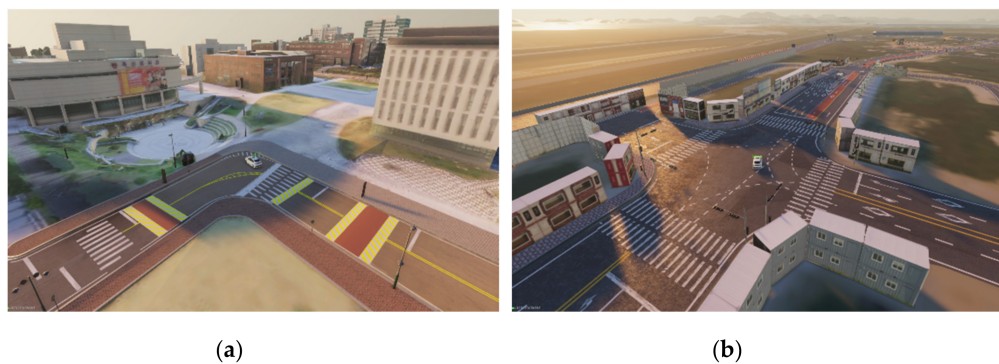

(**a**)          (**b**)

**Figure 1.** MORAI Sim Standard: (**a**) Chungbuk National University map; (**b**) KATRI K-city map.

### 3.1. Data Collection

By utilizing the simulator's characteristics to collect various ground truths, data related to the camera and camera pose were acquired. The process was configured so that a separate handcraft labeling operation was not required.

An RGB camera image for the input of the whole system, the segmentation ground truth image used for the backbone semantic segmentation, and the pose (x, y, z, roll, pitch, and yaw) expressing the camera attachment position were acquired.

Among the pose data, the roll, pitch, and yaw, which express the angles of each 3D axis, ranged from 0 to 360 degrees. Due to the vehicle's windshield, it is difficult to express the rotation of the camera in real vehicle, so only a tiny change in pose compared to the entire range can be expressed. In this paper, various camera poses were constructed using the simulator, because overfitting occurs when learning with data with a small number of configurable data collection groups compared to the actual range.

### 3.2. LUT Generation

Due to the nature of the simulator that provides a camera image without lens system distortion, the distortion removal procedure is omitted, an ideal camera matrix is created, and the rotation and translation matrices are generated using the extracted camera pose roll/pitch/yaw and translation information. $c_x/c_y$ means the principal length of the camera and $f_x/f_y$ means the focal length, and based on these contents, the camera matrix $K$ is estimated, which is a conversion matrix between the camera's original plane and the normalized plane.

$$c_x = width/2$$

$$c_y = height/2$$

$$pi = 3.141592...$$

$$f_x = c_x / \left( \tan \left( 0.5 * fov * \frac{pi}{180} \right) \right)$$

$$f_y = c_y / \left( \tan \left( 0.5 * fov * \frac{pi}{180} \right) \right)$$

$$f = \begin{cases} f_x & (f_x \geq f_y) \\ f_y & (f_x < f_y) \end{cases}$$

$$K = \begin{bmatrix} f & 0 & c_x \\ 0 & f & c_y \\ 0 & 0 & 1 \end{bmatrix}$$

In the homography production stage, through image calibration, the existing methods combine the $3 \times 3$ rotation matrix ($R$) and the $3 \times 1$ translation matrix ($T$) to create and utilize a $3 \times 4$ matrix $RT$, but if this method is used, the axis before rotation is applied. Due to the gimbal lock phenomenon that occurs when huge angles are rotated on an axis, the conversion may not be performed properly. Therefore, we multiplied $3 \times 4$ $R|T_{null}$ first and multiplied $3 \times 4$ $R_i|T$ later to solve the problem. The problem was solved by considering the rotation of the coordinate axis first and then applying a translation matrix based on a new three-dimensional orthogonal axis ($3 \times 3$ unit matrix) rather than the rotated axis (maybe with gimbal lock). The details are as follows.

$$R|T_{null} = \begin{bmatrix} r_{11} & r_{12} & r_{13} & 0 \\ r_{21} & r_{22} & r_{23} & 0 \\ r_{31} & r_{32} & r_{33} & 0 \end{bmatrix}$$

$$R_i|T = \begin{bmatrix} 1 & 0 & 0 & t_x \\ 0 & 1 & 0 & t_y \\ 0 & 0 & 1 & t_z \end{bmatrix}$$

$$RT = R_i|T * R|T_{null}$$

Since the $RT$ obtained in this way is a $3 \times 4$ matrix, which is a transformation between a 3D homogeneous coordinate system and a 2D homogeneous coordinate system, the $z$-axis data are removed, based on the camera coordinates, to be used in the BEV transformation (which is a 2D-to-2D transformation, z = 0). If column 3 is deleted in $RT$, an effect in which the $z$-axis data becomes 0 is derived, and a $3 \times 3$ matrix is obtained.

The homography $H$ between the original image and the BEV is the product of $RT_{3rdColRemoved}$ (which is the rotation matrix between the image and the BEV), the camera matrix $K$, and the matrix that makes the scale and makes left-top of the BEV to $(0, 0)$.

$$H = \begin{bmatrix} bevScale & 0 & -bevWidth/2 \\ 0 & bevScale & -bevHeight \\ 0 & 0 & 1 \end{bmatrix} * K * RT_{3rdColRemoved}$$

Both the $x$ and $y$ coordinates of the original image corresponding to the BEV points can be obtained using homography $H$. The $x$ and $y$ coordinate values are divided by the width and height of the original image, multiplied by 65,525, converted into 16-bit data, and then stored as images named LUT_X and LUT_Y, respectively.

## 4. Proposed Deep Learning Network

The network (Figure 2) consists of an encoder, a decoder, an LUT generator, and a pose regressor. As loss functions, segmentation loss in the encoder, LUT loss in the LUT generator, and pose loss in the pose regressor were used.

The input of the whole network is a three-channel RGB image, and the output is the predicted pose and LUT of the BEV. The predicted pose is output through the encoder and pose regressor, and the LUT is output through the encoder, decoder, and LUT generator.

The pose regressor is attached in the form of an add-on, and by connecting it, the overall direction of the network is given and the performance is improved. The predicted pose helps the user to intuitively monitor the learning progress during the learning process.

The image output in the LUT reduces the post-processing time because it enables the BEV to be produced immediately after performing only the memory copy process and simple interpolation operation without additional operation.

In Tables 1–5, C denotes the customizable channel, and H and W denote the height and width of the input image (or input feature map), respectively.

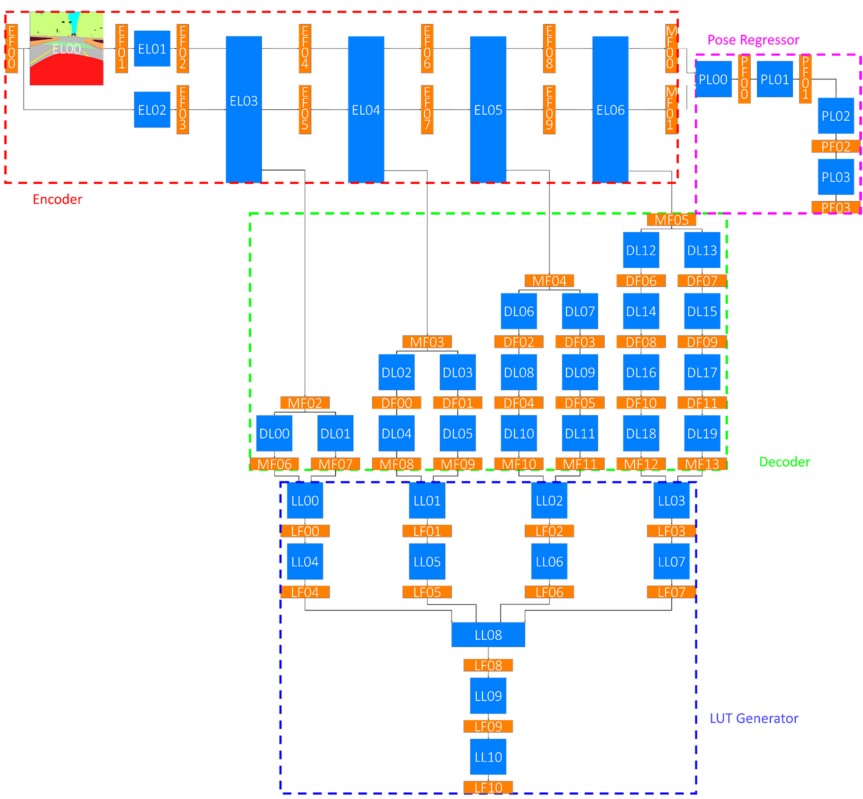

**Figure 2.** Entire network structure.

**Table 1.** (**a**). Encoder layer detail; (**b**). Encoder feature map detail.

| | | **(a)** | | |
|---|---|---|---|---|
| **Name** | **In Channels** | **Out Channels** | **Layer** | **Remarks** |
| EL00 | 3 | C | Semantic segmentation | |
| EL01 | C | 16 | 2D convolution (kernel: 3; stride: 1; padding: 1) | |
| EL02 | 3 | 16 | | Encoder layer |
| EL03 | 16 | 64 | | |
| EL04 | 64 | 256 | Compress block | |
| EL05 | 256 | 1024 | | |
| EL06 | 1024 | 4096 | | |

| | | **(b)** | | |
|---|---|---|---|---|
| **Name** | **Channel** | **Height** | **Width** | **Remarks** |
| EF00 | 3 | H | W | |
| EF01 | C | H | W | |
| EF02 | 16 | H | W | |
| EF03 | | | | |
| EF04 | 64 | H/2 | W/2 | Encoder feature |
| EF05 | | | | |
| EF06 | 256 | H/4 | W/4 | |
| EF07 | | | | |
| EF08 | 1024 | H/8 | W/8 | |
| EF09 | | | | |

**Table 1.** *Cont.*

| | | | | |
|---|---|---|---|---|
| MF00 | 4096 | H/16 | W/16 | Middle feature |
| MF01 | | | | |
| MF02 | 64 | H/2 | W/2 | |
| MF03 | 256 | H/4 | W/4 | |
| MF04 | 1024 | H/8 | W/8 | |
| MF05 | 4096 | H/16 | W/16 | |

**Table 2.** (**a**). Compress Block Layer Detail; (**b**). Compress block feature map detail. (**c**). Custom residual block layer detail. (**d**). Custom residual block feature map detail.

| (a) | | | | |
|---|---|---|---|---|
| **Name** | **In Channels** | **Out Channels** | **Layer** | **Remarks** |
| CL00 | C | 4C | Custom residual block | Compress layer |
| CL01 | | | | |
| CL02 | 4C | 8C | Concatenate | |
| CL03 | 8C | 4C | 2D convolution (kernel: 3; stride: 1; padding: 1) | |

| (b) | | | | |
|---|---|---|---|---|
| **Name** | **Channel** | **Height** | **Width** | **Remarks** |
| CF00 | C | H | W | Compress feature |
| CF01 | | | | |
| CF02 | 4C | H/2 | W/2 | |
| CF03 | | | | |
| CF04 | 8C | H/2 | W/2 | |
| CF05 | 4C | H/2 | W/2 | |
| CF06 | | | | |
| CF07 | | | | |

| (c) | | | | |
|---|---|---|---|---|
| **Name** | **In Channels** | **Out Channels** | **Layer** | **Remarks** |
| RL00 | C | 2C | 2D convolution (kernel: 3; stride: 2; padding: 1) | Residual layer |
| RL01 | 2C | 4C | 2D convolution (kernel: 3; stride: 1; padding: 1) | |
| RL02 | C | 4C | 2D convolution (kernel: 3; stride: 2; padding: 1) | |
| RL03 | 4C | 4C | Element Add | |

| (d) | | | | |
|---|---|---|---|---|
| **Name** | **Channel** | **Height** | **Width** | **Remarks** |
| RF00 | C | H | W | Residual feature |
| RF01 | 2C | H/2 | W/2 | |
| RF02 | 4C | H/2 | W/2 | |
| RF03 | 4C | H/2 | W/2 | |
| RF04 | 4C | H/2 | W/2 | |

**Table 3.** (**a**). Decoder layer detail. (**b**). Decoder feature map detail.

| (a) | | | | |
|---|---|---|---|---|
| **Name** | **In Channels** | **Out Channels** | **Layer** | **Remarks** |
| DL00 | 64 | 16 | Pixel shuffle | Decoder layer |
| DL01 | 64 | 16 | Convolution transposed (kernel: 2; stride: 2; padding: 0) | |
| DL02 | 256 | 64 | Pixel shuffle | |

**Table 3.** *Cont.*

| DL03 | 256 | 64 | Convolution transposed (kernel: 2; stride: 2; padding: 0) |
|------|-----|----|----------------------------------------------------------|
| DL04 | 64 | 16 | Pixel shuffle |
| DL05 | 64 | 16 | Convolution transposed (kernel: 2; stride: 2; padding: 0) |
| DL06 | 1024 | 256 | Pixel shuffle |
| DL07 | 1024 | 256 | Convolution transposed (kernel: 2; stride: 2; padding: 0) |
| DL08 | 256 | 64 | Pixel shuffle |
| DL09 | 256 | 64 | Convolution transposed (kernel: 2; stride: 2; padding: 0) |
| DL10 | 64 | 16 | Pixel shuffle |
| DL11 | 64 | 16 | Convolution transposed (kernel: 2; stride: 2; padding: 0) |
| DL12 | 4096 | 1024 | Pixel shuffle |
| DL13 | 4096 | 1024 | Convolution transposed (kernel: 2; stride: 2; padding: 0) |
| DL14 | 1024 | 256 | Pixel shuffle |
| DL15 | 1024 | 256 | Convolution transposed (kernel: 2; stride: 2; padding: 0) |
| DL16 | 256 | 64 | Pixel shuffle |
| DL17 | 256 | 64 | Convolution transposed (kernel: 2; stride: 2; padding: 0) |
| DL18 | 64 | 16 | Pixel shuffle |
| DL19 | 64 | 16 | Convolution transposed (kernel: 2; stride: 2; padding: 0) |

**(b)**

| Name | Channel | Height | Width | Remarks |
|------|---------|--------|-------|---------|
| DF00 | | | | |
| DF01 | 64 | H/2 | W/2 | |
| DF02 | | | | |
| DF03 | 256 | H/4 | W/4 | |
| DF04 | | | | |
| DF05 | 64 | H/2 | W/2 | |
| DF06 | | | | Decoder feature |
| DF07 | 1024 | H/8 | W/8 | |
| DF08 | | | | |
| DF09 | 256 | H/4 | W/4 | |
| DF10 | | | | |
| DF11 | 64 | H/2 | W/2 | |
| MF02 | 64 | H/2 | W/2 | |
| MF03 | 256 | H/4 | W/4 | |
| MF04 | 1024 | H/8 | W/8 | |
| MF05 | 4096 | H/16 | W/16 | |
| MF06 | | | | |
| MF07 | | | | Middle feature |
| MF08 | | | | |
| MF09 | | | | |
| MF10 | 16 | H | W | |
| MF11 | | | | |
| MF12 | | | | |
| MF13 | | | | |

**Table 4.** (**a**). LUT generator layer detail. (**b**). LUT generator feature map detail.

| (a) | | | | |
|---|---|---|---|---|
| Name | In Channels | Out Channels | Layer | Remarks |
| LL00 | | | | |
| LL01 | 16 | 32 | Concatenate | |
| LL02 | | | | |
| LL03 | | | | |
| LL04 | | | | |
| LL05 | 32 | 16 | 2D convolution (kernel: 3; stride: 1; padding: 1) | LUT layer |
| LL06 | | | | |
| LL07 | | | | |
| LL08 | 16 | 64 | Concatenate | |
| LL09 | 64 | 16 | 2D convolution (kernel: 3; stride: 1; padding: 1) | |
| LL10 | 16 | 3 | 2D convolution (kernel: 3; stride: 1; padding: 1) | |
| (b) | | | | |
| Name | Channel | Height | Width | Remarks |
| LF00 | | | | |
| LF01 | 32 | H | W | |
| LF02 | | | | |
| LF03 | | | | |
| LF04 | | | | |
| LF05 | 16 | H | W | LUT feature |
| LF06 | | | | |
| LF07 | | | | |
| LF08 | 64 | H | W | |
| LF09 | 16 | H | W | |
| LF10 | 3 | H | W | |
| MF06 | | | | |
| MF07 | | | | |
| MF08 | | | | |
| MF09 | | | | |
| MF10 | 16 | H | W | Middle feature |
| MF11 | | | | |
| MF12 | | | | |
| MF13 | | | | |

**Table 5.** (**a**). Pose regressor layer detail. (**b**). Pose regressor feature map detail.

| (a) | | | | |
|---|---|---|---|---|
| Name | In Channels | Out Channels | Layer | Remarks |
| PL00 | 4096 | 8192 | Concatenate | |
| PL01 | 8192 | 1024 | 2D convolution (Kernel: 3; stride: 1; padding: 1) | Pose layer |
| PL02 | 1024 | 4HW | Flatten (3D to 1D) | |
| PL03 | 4HW | 3 | Fully connected layer | |

**Table 5.** *Cont.*

|  | (b) |  |  |  |
|---|---|---|---|---|
| **Name** | **Channel** | **Height** | **Width** | **Remarks** |
| PF00 | 8192 | H/16 | W/16 | |
| PF01 | 1024 | H/16 | W/16 | |
| PF02 | 4HW | 1 | 1 | Pose Feature |
| PF03 | 3 | 1 | 1 | |
| MF00 | 4096 | H/16 | W/16 | |
| MF01 | | | | |

### 4.1. Encoder

To create an LUT that converts to a BEV using a single camera, we used semantic segmentation-based features [5] for the semantic segmentation backbone network. After fitting the original image and the segmentation result to the same scale, down-sampling is performed through the compress block. The output from each compress block comprises encoded data for multiple scales, and through this, a network robust to multiple scales is constructed.

The semantic segmentation layer aims to find precise boundaries for objects in the image. In addition, since the proposed network aims to estimate the pose using image features, we inserted a semantic segmentation layer into the encoder (Figure 3, Table 1) to utilize the information on the precise boundary as a feature.

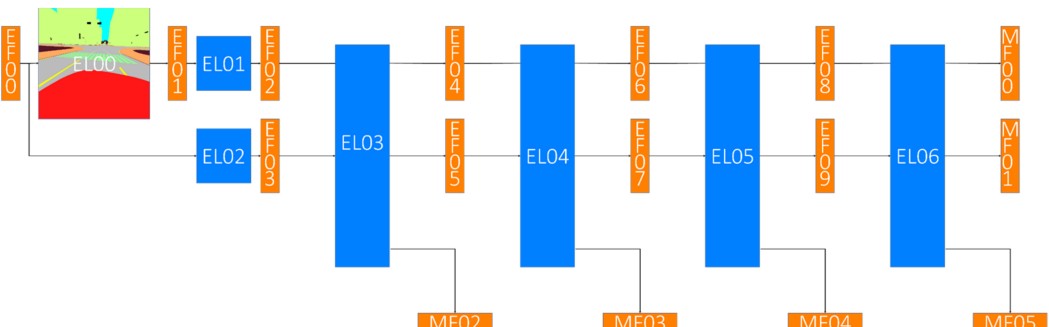

**Figure 3.** Encoder structure.

The compress block (Figure 4a, Table 2a,b) plays a role in integrating features using the custom residual block (Figure 4b, Table 2c,d), referring to He et al. [17], and the structure that fuses the original image and segmentation features.

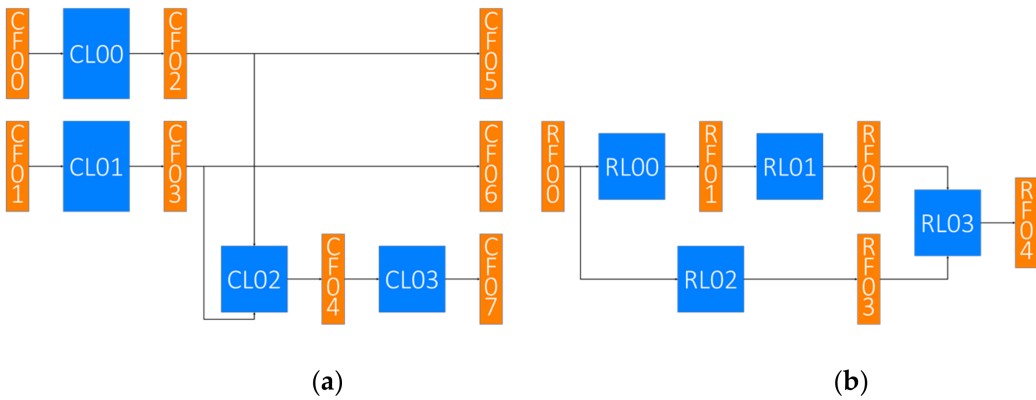

**Figure 4.** (**a**) Compress block structure; (**b**) custom residual block structure.

### 4.2. Decoder

Due to the characteristics of a camera that projects light in a specific space, far-distance data are insufficient compared to near-distance data, which cause aliasing. We constructed a parallel path to efficiently perform anti-aliasing by utilizing the features delivered from the encoder.

A structure for restoring and up-scaling multi-scale encoded data was constructed by composing a parallel path through the transposed convolution [8] and pixel shuffle [18] to solve the aliasing phenomenon that may occur in the up-scaling process.

The decoder (Figure 5, Table 3) does not output a separate loss, but delivers data to the LUT generator.

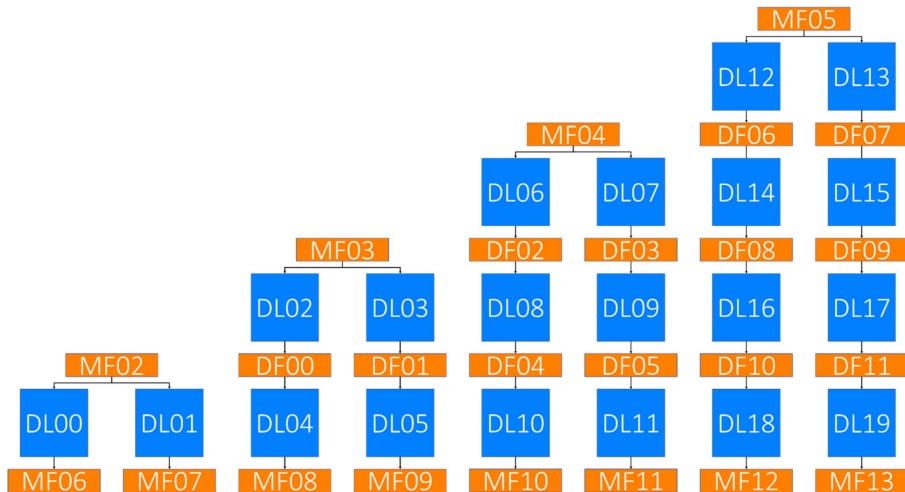

**Figure 5.** Decoder structure.

### 4.3. LUT Generator

Through converging and compressing the results obtained through the decoder, three LUT channels were finally generated. The first/second channels represent x/y coordinates of the original image, respectively, and the third channel represents the boundary for the camera's field of view (FoV) area during the LUT conversion process (the boundary value is the max value, while the rest is the min value). Figure 6 and Table 4 show the structure of the LUT generator.

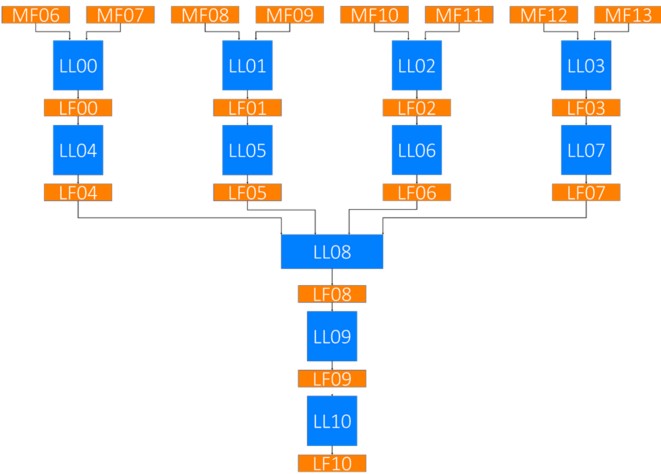

**Figure 6.** LUT generator structure.

### 4.4. Pose Regressor

In essence, the LUT is an interpretation of the geometrical information of the image (the geometrical information described in this paper is the camera pose, that is, the roll/pitch/yaw), so we must consider the camera pose regression from the network design stage. To effectively add pose regression information to the network, we aimed to make the network recognize the task of estimating the pose by attaching a pose regressor (Figure 7, Table 5) to the network.

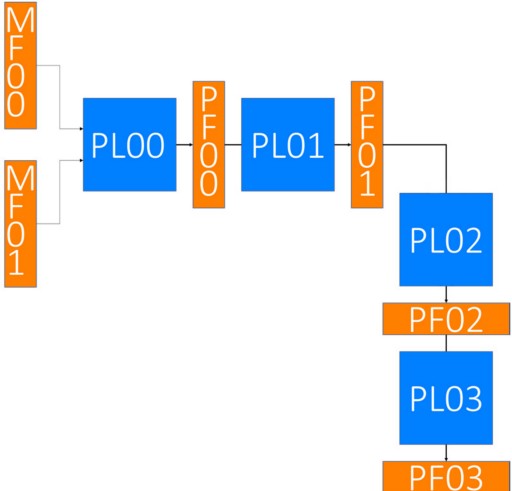

**Figure 7.** Pose regressor structure.

To estimate the pose, we compressed the encoder data and utilized the fully connected layer [19] to understand the correspondence of all the information between the encoder features [12].

In the angle expression method using the degree or radian, the non-continuous parts such as 0 degrees/360 degrees and –π/π may hinder the learning performance, so cos is used and then translated, while the region of the value is scaled to change from –1~1 to 0~1, and the activation function of the final layer is used as a sigmoid to efficiently infer the value of 0~1.

## 5. Experiments

### 5.1. Loss Cost

Three feature points calculate loss cost. The seg loss is at the end of the segmentation backbone of the encoder, the LUT loss is located at in LUT generator output, and the pose loss is obtained from the pose regressor output.

The contour of the semantic segmentation feature must be precise to help improve the entire network's performance, so we output the segmentation loss using the pixel-wise cross-entropy [20] of the segmentation.

The LUT loss is calculated through pixel-wise mean squared error (MSE), and the weights of the first/second and third channels, which have different basic properties, are experimentally learned differently.

For the result of the pose regressor, the pose loss is obtained by using the MSE. Due to the relatively small number of elements, a smaller value is output compared to the other losses.

Since the domain covered by each loss and the convergence speed in the learning process are different, the weight multiplied by each loss cost in calculating the total loss was experimentally obtained and is as follows.

$$SegLoss = CrossEntropy(Seg)$$

$$LUTLoss = 5 * MSE(LUTX) + 5 * MSE(LUTY) + 3 * MSE(LUTFOV)$$

$$PoseLoss = MSE(Pose)$$

$$TotalLoss = 1 * SegLoss + 5 * LUTLoss + 100 * PoseLoss$$

### 5.2. Quantitative Evaluation

In this paper, the seg loss is a learning measure for segmentation features inside the network, and the LUT and pose losses are quantitative indicators for the LUT/3D spatial angle, respectively (Table 6).

**Table 6.** Differences between the versions of the network (v1~v4).

| No | Scale | Parallel Path | Pose Regressor | Processing Time (s) | Seg Loss | LUT Loss | Pose Loss |
|----|-------|---------------|----------------|---------------------|----------|----------|-----------|
| v1 | Single | 2 pixel shuffles | X | 0.18 | 0.65 | 1.59 | - |
| v2 | Single | Pixel shuffle and convolution transposed | X | 0.20 | 0.58 | 0.09 | - |
| v3 | Multi | Pixel shuffle and convolution transposed | X | 0.74 | 0.22 | 0.04 | - |
| v4 | Multi | Pixel shuffle and convolution transposed | O | 0.81 | 0.14 | 0.01 | 0.14 |

We gradually changed the layers to infer the change in performance from the structural evolution of our network (Table 7).

**Table 7.** Metrics of gradual change for the network.

| From | To | Processing Time Change (To/From) | Seg Loss Change (To/From) | LUT Loss Change (To/From) |
|------|-----|----------------------------------|---------------------------|---------------------------|
| v1 | v2 | 1.11 | 0.89 | 0.06 |
| v2 | v3 | 3.70 | 0.38 | 0.44 |
| v3 | v4 | 1.09 | 0.64 | 0.25 |

The resolution representation of the encoder, decoder, and LUT generator was tested with 1 (single scale)/4 (multi-scale), and the parallel path of the decoder was tested with two pixel shuffles or pixel shuffle and convolution transposed. Finally, we tried to improve the performance through the combination with a pose regressor.

All tests were inferenced in the NVIDIA GTX 1080 environment, and three losses were evaluated.

If the parallel path composed of two pixel shuffles is changed to a layer consisting of pixel shuffle and convolution transposed, the speed is 1.11 times slower, but the segmentation loss is reduced 0.89 times and the LUT loss is significantly reduced as well (0.06 times).

If the resolution representation considered by the encoder, decoder, and LUT generator is changed from single scale to multi-scale, the speed is 3.7 times slower, the seg-mentation loss is 0.38 times less, and the LUT loss is 0.44 times less, so we can see that the loss decreases. When changing to multi-scale, we can see that the segmentation loss is significantly reduced.

When learning that by adding a pose regressor to the end of the encoder, the speed is only reduced 1.09 times, but the segmentation loss is reduced 0.64 times and the LUT loss by 0.25 times, we can see that the loss cost is significantly reduced, while the processing speed is slightly increased. Through this, we can understand that it is meaningful to connect the pose regressor to the end of the encoder to obtain the direction of the entire network.

### 5.3. Qualitative Evaluation

Since the coordinates of the original image corresponding to each coordinate of the BEV can be estimated using the LUT data obtained from the end of the network, the BEV

was generated through this value. It was tested using two map data of Chungbuk National University (CBNU) and KATRI K-city (Tables 8 and 9).

**Table 8.** Chungbuk National University map-based BEV generation evaluation.

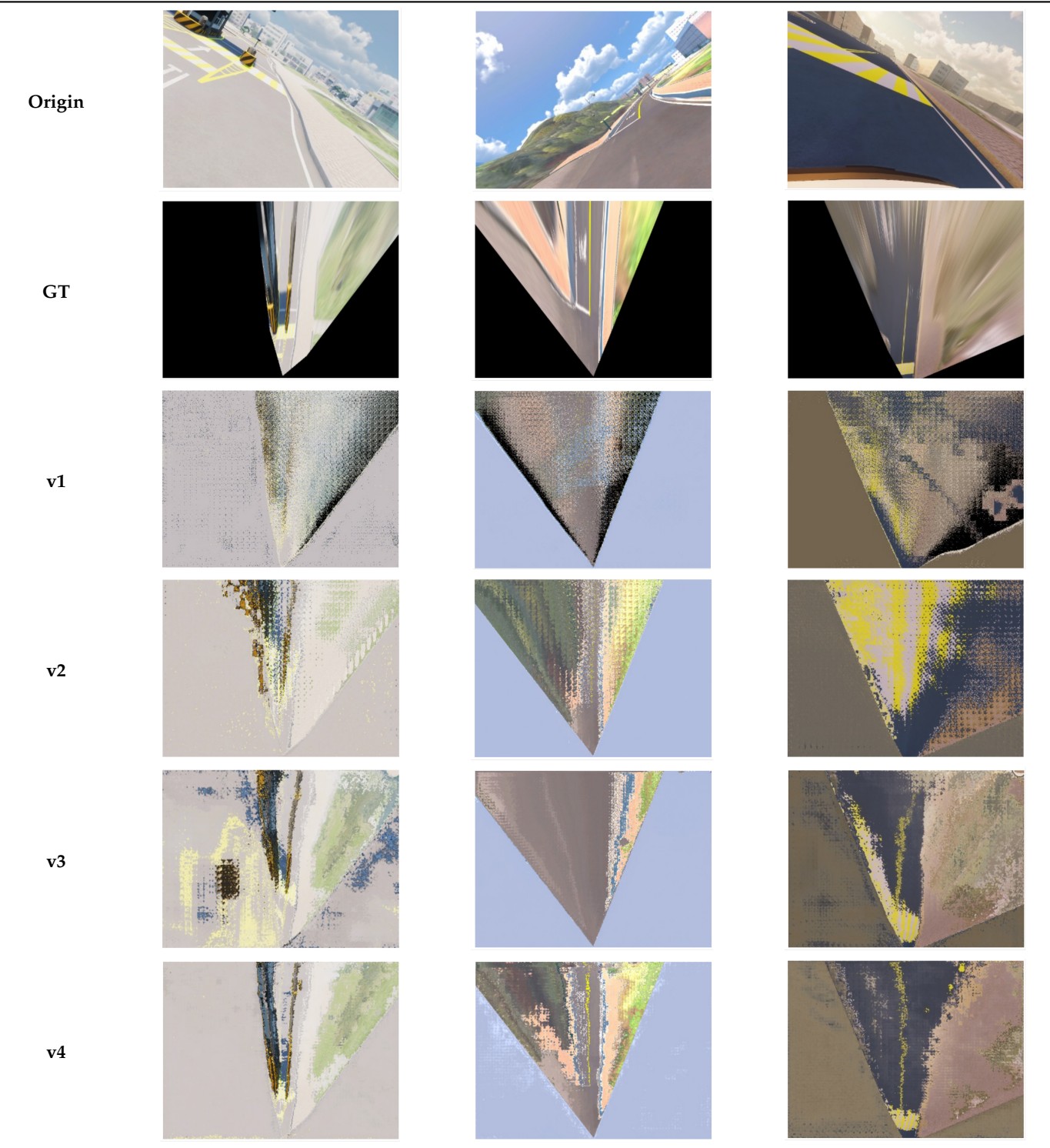

**Table 9.** KATRI K-city map-based BEV generation evaluation.

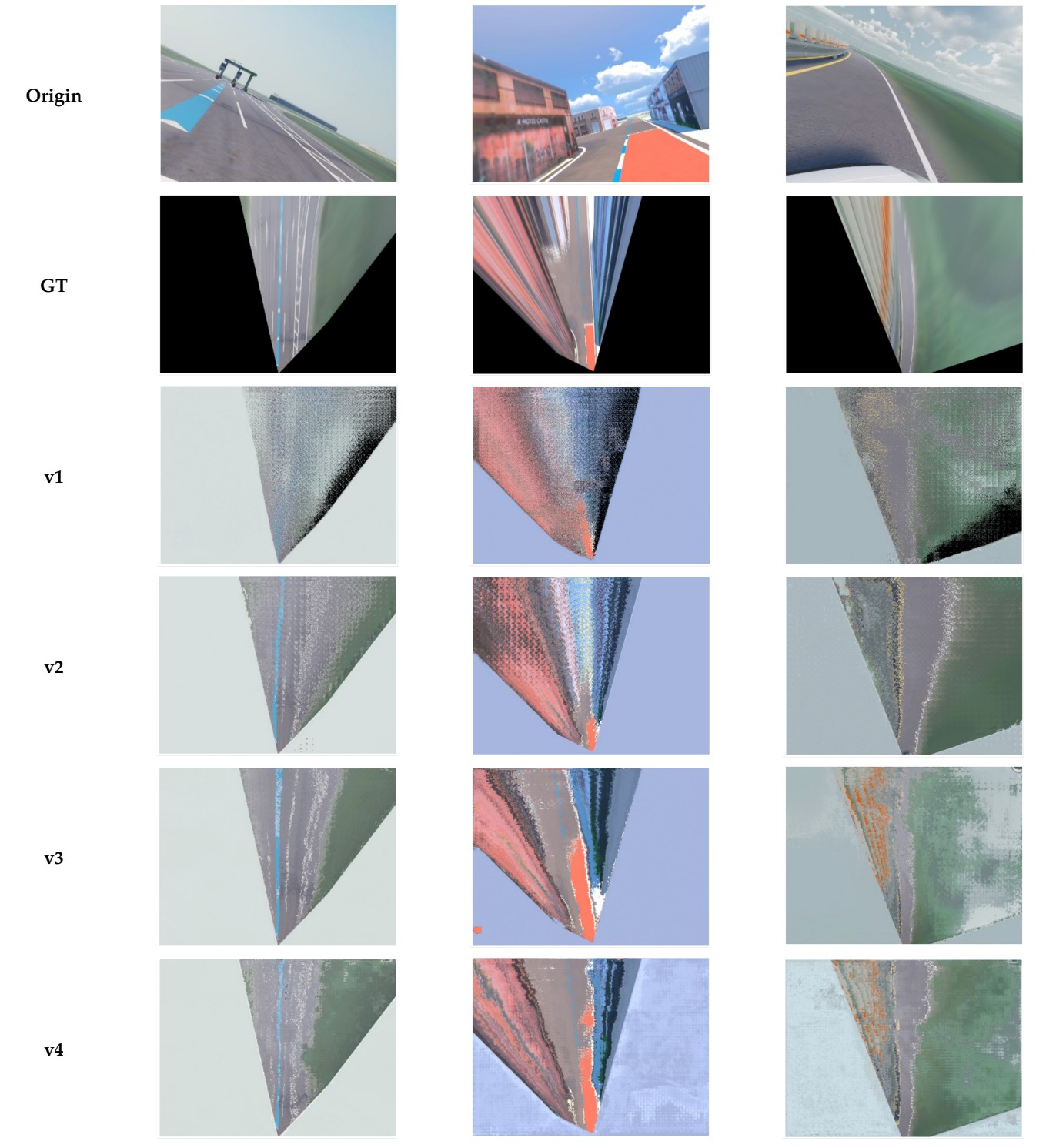

As we progressed from v1 to v4, the aliasing decreased. Particularly, if we compare v3 (without a pose regressor) and v4 (with a pose regressor), we can see that the concept of the overall pose is added, and the distant region is converted relatively well.

## 6. Conclusions

In this work, we studied BEV conversion based on a single camera image. We used segmentation backbone-based features during the study, and the performance difference

before and after attachment was analyzed by adding on a pose regressor. Since it is challenging to collect various camera poses using an actual camera, we tested the network through a simulator.

We plan to conduct research using actual camera data (or actual + synthetic data) in this network in the future and try to reduce aliasing by improving the network. In addition, to supplement the characteristics of a single camera, which makes it difficult to estimate scale, it is intended to produce a real distance-based BEV with an explicit unit rather than a relatively real distance through combination with a ToF sensor or other odometry methods, as with adjacent frames of a single camera.

**Author Contributions:** Conceptualization, D.L., W.P.T.; methodology, D.L., W.P.T.; validation, D.L.; writing—original draft preparation, D.L.; writing—review and editing, S.-C.K.; supervision, S.-C.K.; funding acquisition, S.-C.K. All authors have read and agreed to the published version of the manuscript.

**Funding:** This research was supported by the MOTIE (Ministry of Trade, Industry, and Energy) in Korea, under the Fostering Global Talents for Innovative Growth Program (P0008751) supervised by the Korea Institute for Advancement of Technology (KIAT). This research was supported by the MSIT (Ministry of Science and ICT), Korea, under the Grand Information Technology Research Center support program (IITP-2021-2020-0-01462) supervised by the IITP (Institute for Information & communications Technology Planning & Evaluation.

**Institutional Review Board Statement:** Not applicable.

**Informed Consent Statement:** Not applicable.

**Data Availability Statement:** Not applicable.

**Conflicts of Interest:** The authors declare no conflict of interest.

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
