# Peer review of "Birds Eye View Look-Up Table Estimation with Semantic Segmentation"

_applsci, doi:10.3390/app11178047_

Round 1
Reviewer 1 Report
Dear Authors!
The topic of the paper is interesting, but quality of presentation is poor, that makes paper difficult to read.
Major issues:
- Text in Figs. 4-7 is unreadable
- All variables in equations should be described
- Please let a native speaker read the text.
Some minor issues:
- Line 16: "Most" should start with a small letter
- Line 16: insert space between ToF and (
- Line 37: "network work"
- Line 49: "BEV was used as a method" - BEV is not a method
- Line 51: "Obstacle" should start with a small letter
- Line 66: What is DB? Database?
Reviewer 2 Report
The paper is very difficult to read because it is full of typos and sentences that make no sense to me, sometimes because of the sentence itself and sometimes because some part is repeated or the sentence is incomplete.
It is not a good practice to use an acronym as a section title (section 3).
The abstract is not a good place to include citations, because it is just an overview of what is developed in the paper. This abstract seems to be part of the introduction.
When referencing a work it is common practice to name the authors, not the title of the paper, as its common practice in section 2.
Figures 2 to 7 cannot be read when printed.
From the point of view of the contents, I think that the explanation about how the rigid transformation of the pose it is a bit strange. Just explaining which ones are the reference systems used simplify all the explanation. A rigid transformation with a projection along the Z axis is [R | t], where R is the rotation matrix that aligns the initial reference system with the destination one, and t is the position of the origin of the initial reference system expressed in coordinates of the destination one.
I really do not understand the word "semantic segmentation" in the title or in the explanations. May be you are making a reference to the fact that your network computes the boundary of the image FOV in the BEV. At first, I expected a segmentation of points that belong to ground plane or not.
From the point of view of the results, they seam just qualitative. As you claim that the network estimates the camera pose, it would be nice to have some quantitative information about the results.
BEV images are not sharp, they present a lot of aliasing. Some discussion about why it happens and it relevance would be interesting.
I also would appreciate a better explanation about image coordinates computation, because the homography matrix produces homogeneous coordinates rather than Cartesian ones.
Round 2
Reviewer 1 Report
-